# Pathogens Detected in Questing *Ixodes ricinus* Ticks in a Mountainous Area in Greece

**DOI:** 10.3390/pathogens13060449

**Published:** 2024-05-25

**Authors:** Katerina Tsioka, Anastasios Saratsis, Styliani Pappa, Anna Papa

**Affiliations:** 1Department of Microbiology, Medical School, Aristotle University of Thessaloniki, 54124 Thessaloniki, Greece; aik.tsioka@gmail.com (K.T.); s_pappa@hotmail.com (S.P.); 2Hellenic Agricultural Organisation-Dimitra (ELGO-DIMITRA), Veterinary Research Institute, 57001 Thermi, Greece; saratsis@elgo.gr

**Keywords:** *Ixodes ricinus*, tick-borne pathogens, *Rickettsia*, *Borrelia*, Greece

## Abstract

*Ixodes ricinus* ticks are vectors of a plethora of pathogens. The purpose of this study was to screen 398 *I. ricinus* ticks for a variety of pathogens. Following the pooling, homogenization, and extraction of total nucleic acids, a real-time PCR was applied for the detection of a panel of tick-borne pathogens, while additional conventional PCRs combined with Sanger sequencing were applied for the detection of viruses and typing of *Rickettsia* and *Borrelia* species. At least one pathogen was detected in 60 of the 80 (75%) tick pools. *Rickettsia* spp. predominated, as it was detected in 63.75% of the pools (51/80; MIR 12.81%), followed by *Borrelia* spp. (35 pools (45%); MIR 8.79%), while *Anaplasma phagocytophilum* was detected in 2 pools (2.5%, MIR 0.5%). The ticks of six *Rickettsia*-positive pools were tested individually (from stored half-ticks); all sequences were identical to those of *R. monacensis*. Similarly, the ticks of six *Borrelia*-positive pools were tested individually, and it was shown that four belonged to the genospecies *Borrelia garinii* and two to *Borrelia valaisiana*. Phleboviruses were detected in 3 pools (3.75%; MIR 0.75%), with sequences clustering in the *Ixovirus* genus, while nairoviruses were detected in 7 pools (8.75%; MIR 1.76%), with one sequence clustering in the *Orthonairovirus* genus, and six clustering in the *Norwavirus* genus. Although a small number of ticks from only one area in Greece were tested, a variety of pathogens together with recently identified viruses were detected, prompting further studies in ticks and surveillance studies in humans.

## 1. Introduction

*Ixodes ricinus* is a widespread tick species in Europe and a known vector for a wide range of pathogens, including *Borrelia*, *Anaplasma*, *Rickettsia*, and *Babesia* species, as well as tick-borne encephalitis virus (TBEV). Ticks of this species undergo a three-host life cycle and primarily inhabit wooded and forested areas where they feed on various mammalian and avian hosts [1]. The distribution of *I. ricinus* is shaped by multiple factors, like environmental and vegetation patterns, and the availability of suitable hosts [2]. The likelihood of pathogen transmission to humans and animals through tick bites is associated with a high density and prevalence of infected ticks alongside hosts’ activities [3].

Several studies have been conducted in Greece aiming to identify tick species and/or detect pathogens in ticks removed from humans or livestock (sheep, goats, and cattle). The results from a study of 519 ticks removed from humans showed that 81.5% were *Rhipicephalus sanguineus*, while *I. ricinus* accounted for only 1% [4]. The tick prevalence and distribution vary among studies on ticks collected from domestic animals depending on the location, animal host, and seasonality. As an example, a study on 11,620 ticks collected during 1983–1986 from domestic animals in the Macedonia Region of Greece identified 18 tick species and subspecies, with *Rhipicephalus bursa* being the most common tick (36.3%). *I. ricinus* accounted for 6.8% of the ticks and was found mainly during autumn and winter in the biotopes of the attenuated mesomediterranean and submediterranean bioclimates and especially in deciduous forest areas [5]. Similarly, another study on 2108 ixodid ticks collected from small ruminants in Greece showed that *Rhipicephalus sanguineus* s.l. and *Rhipicephalus bursa* were the most prevalent tick species (64.8% and 25.9%, respectively), whereas the least frequently collected species were *I. ricinus*, *Haemaphysalis punctata*, and *Hyalomma impeltatum* (0.04% each) [6]. However, different results were taken from a targeted study in northern Greece, where the collection was performed only during April–July and September–December, and only on livestock (mainly goats) grazing in hilly areas neighboring forests of deciduous and evergreen vegetation, where the average humidity was approximately 80%. In that study, 3249 adult ticks were collected from domestic animals, and *I. ricinus* was the predominant tick species (44.57%) [7]. It has to be mentioned that there is a difference between collecting ticks from hosts and collecting ticks directly from vegetation, as in the first case, the ticks have already fed and are more likely to be infected, in contrast with questing ticks. To the best of our knowledge, there is not any study on pathogen testing in host-seeking ticks in Greece.

The application of 16S rRNA next-generation sequencing in ticks belonging to four genera (*Ixodes*, *Rhipicephalus*, *Dermacentor*, and *Haemaphysalis*), which were collected from goats, showed that the predominant phylum in *I. ricinus* ticks was Proteobacteria, with bacteria of the *Rickettsiaceae* (*Rickettsia* and *Anaplasma* species) and “*Candidatus* Midichloriaceae” families being detected [8]. Another small-scale study showed that 66.6% and 33.3% pools of *I. ricinus* ticks collected from goats were positive for *Rickettsia monacensis* and *Anaplasma platys*, respectively [9].

Regarding viruses, Greek goat encephalitis virus (GGEV) from the tick-borne encephalitis (TBE) group, was isolated in 1969 from the brain of a newborn goat with neurological symptoms [10]. To identify the tick vector of GGEV, *Ixodidae* ticks collected from goats and sheep in rural areas of northern Greece were tested, and GGEV was detected in two pools of *I. ricinus* ticks [7]. The virus was then isolated from one of the positive pools [11]. A few human TBE cases have been reported in Greece during the last years (article in preparation), suggesting that the virus is present in the country and needs further attention and surveillance studies.

To gain a better insight into the pathogens carried by *I. ricinus* ticks in Greece, a mountainous area with an ecosystem favorable for this tick species was selected for tick collection and screening for various pathogens. To our knowledge, this is the first study focusing on the pathogen analysis of questing *I. ricinus* ticks in Greece.

## 2. Materials and Methods

### 2.1. Study Area

Mount Vermion (40°31′32.9″ N 22°00′37.4″ E) is located in the central part of northern Greece, having a north–south orientation. The study area included 83 locations in the municipalities of Veria, Naoussa, and Edessa (Figure 1). The collective population of the area is 199,116 according to the most recent census data [12]. The slopes facing to the east are influenced by pluvial aerial masses coming from the Aegean Sea; as a result, the ecosystem is more productive than the west-facing sites. The mountain is bounded to the east by the Central Macedonian plain, characterized by intensive cultivation of fruit-bearing trees. Its southern and northern boundaries are delineated by the Aliakmonas and Agras rivers, respectively. During the last decades, the gradual abandonment of transhumance in the highlands of the mountain has resulted in a relative increase in forested areas. In addition, the area is undergoing considerable land use changes, marked by the implementation of renewable energy projects, such as wind farms and solar parks. The climate of the area is temperate Mediterranean, characterized by rainy winters and warm summers. The total annual rainfall in the area is 1500 mm, whereas the mean annual precipitation is approximately 699.83 mm, and the mean annual air temperature is 15 °C [13]. The minimum rainfall occurs during July–August; however, the atmosphere remains humid due to its proximity to the archipelagos.

### 2.2. Sampling

From April 2021 to March 2022, drag sampling was conducted by using a 1 m^2^ flannel cloth. An attempt was made to revisit the sites surveyed in the spring and summer of 2021, with subsequent samplings extending into the autumn, winter, and spring of 2021 and 2022, resulting in a cumulative total of 135 sampling occurrences. The sampling was conducted in a diverse range of ecological niches, including deciduous forests, characterized mainly by *Carpinus orientalis*, *Ostrya carpinifolia*, *Quercus frainetto*, *Q. pubescens*, *Q. trojana*, *Castanea sativa*, *Tilia tomentosa*, and *Fagus sylvatica*, occurring individually or in various combinations. Furthermore, the investigations were extended to coniferous evergreen forested areas, where *Pinus nigra* predominated, and to typical maquis vegetation, where the dominant shrub species was *Quercus coccifera*, followed by *Buxus sempervirens*. The 83 survey locations encompassed a spectrum of settings, including forested sites adjacent to agricultural and grazing zones, local recreational areas, hiking trails, hunting grounds, and areas designated under the NATURA 2000 conservation program within the mountainous terrain. The altitudinal range for the samplings spanned from 65 to 1900 m above sea level. Altitudinal data and coordinates were recorded utilizing the SW Maps application (version 2.10.1.0). A sampling location map (Figure 1) was generated employing QGIS (version 3.16.14).

### 2.3. Tick Identification and Transportation

The identification of ticks was performed under a stereomicroscope on an ice pack using morphological identification keys [14]. Then, they were transferred to the laboratory in dry ice and kept at −80 °C until processing.

### 2.4. Nucleic Acid Isolation and Pathogen Detection

Adult ticks were washed with distilled water and vertically cut in half. One-half of each tick was stored in an RNA stabilization reagent (Qiagen, Hilden, Germany) at −80 °C, and the other half was used for the pool preparation in phosphate-buffered saline (2–9 half-ticks per pool) based on the date and site of collection. The same procedure, but without cutting, was followed for nymphs (3–17 per pool) and larvae (one pool of 2). The pooled ticks were homogenized via vortexing for 5 min using glass beads (with a diameter of 150–212 μm). Total nucleic acids were extracted from 200 μL of the supernatants using the IndiSpin Pathogen Kit (formerly known as the QIAamp cador Pathogen Mini Kit, INDICAL Bioscience GmbH, Leipzig, Germany) according to the manufacturer’s instructions. The CerTest VIASURE Tick-Borne Diseases Real-Time PCR Detection Kit (Abacus dx, Brisbane, Australia) was used for the detection of TBEV (the 3′UTR), *Rickettsia* spp. (23S rRNA gene), *Babesia microti* (CCT-eta gene), *Babesia divergens* (hsp70 gene), *Ehrlichia chafeensis* and *Ehrlichia muris* (GroEl gene), *Borrelia burgdorferi* sensu lato (s.l.), *Borrelia miyamotoi* and/or *Borrelia hermsii* (23S rRNA gene), *Anaplasma phagocytophilum* (msp2 gene), and *Coxiella burnetii* (IS1111 gene).

The minimum infection rate (MIR) and maximum likelihood estimation (MLE) with 95% confidence intervals (CIs) for unequal pool sizes were calculated using the PooledInfRate program version 4.0 available at https://www.cdc.gov/mosquitoes/php/toolkit/mosquito-surveillance-software.html accessed on 6 May 2024 [15]. For each pathogen, the respective MIR and MLE values were calculated per 100 ticks tested.

To identify *Borrelia* spp. and *Rickettsia* spp. at the genospecies level, ticks of the *Rickettsia* and *Borrelia* positive pools with Ct values of <28 were further tested individually (the stored half-ticks) using in-house PCR assays, which amplified a 300 bp fragment of the 5S-23S rRNA intergenic spacer of *B. burgdoferi* s.l. [16] and a 381 bp fragment of the citrate synthase gene of *Rickettsia* spp. [17]. Additional in-house PCR assays were applied for the amplification of partial segments of the NS5 gene of flaviviruses [18] and polymerase genes of phleboviruses (the *Phenuiviridae* family) [19] and nairoviruses (the *Nairoviridae* family) [20].

### 2.5. Sanger Sequencing and Phylogenetic Analysis

The PCR products of the in-house PCRs were Sanger-sequenced in a 3130 ABI Genetic Analyzer (Applied Biosystems, Foster City, CA, USA). The nucleotide sequences were compared with respective ones available in GenBank using the National Center for Biotechnology Information (NCBI) Basic Local Alignment Sequence Tool (BLAST) search engine (https://blast.ncbi.nlm.nih.gov/, accessed on 5 February 2024) to identify the best match. Viral sequences were aligned with respective ones retrieved from the GenBank Database using Clustal W2, and evolutionary distances were computed using MEGA version 11 [21]. Maximum likelihood phylogenetic trees were constructed based on the best-fitted nucleotide substitution model.

## 3. Results

### 3.1. Ticks

A total of 440 ticks were collected, and 398 (90.45%) were identified as *I. ricinus*; most of them (87.2%) were adults (Table 1). They were found in 47.4% (64/135) of the samplings, at least once in 51.8% (43/83) of the sampled locations, mainly in deciduous forests (all developmental stages), but occasionally also in typical maquis vegetation (only adults). No ticks were present in *Pinus nigra* forested areas. Adult *I. ricinus* ticks were the predominant developmental stage, irrespective of the period of the year. The altitudinal activity pattern of *I. ricinus* varied from 295 m to 1580 m; it was most commonly present above 600 m.

### 3.2. Pathogen Detection

At least one pathogen was detected in 60 of the 80 (75%) tick pools. *Rickettsia* spp. predominated, as it was detected in 63.75% of the tick pools (51/80). *Borrelia* spp. was detected in 36 pools (45%), *A. phagocytophilum* in 2 (2.5%), phleboviruses in 3 (3.75%), and nairoviruses in 7 (8.75%) (Table 2). The MIR and MLE values are seen in Table 2. Flaviviruses including TBEV, *Babesia microti*, *Babesia divergens*, *Ehrlichia chafeensis* and *Ehrlichia muris*, *Borrelia miyamotoi* and/or *Borrelia hermsii*, and *Coxiella burnetii* were not detected.

Co-infections were detected in 30 (37.5%) tick pools with a variety of combinations. The most common (26.25%) was that of *Rickettsia* and *Borrelia* species (Table 3).

Further analysis through individually tick testing from the six positive pools revealed that all *Rickettsia* sequences presented 100% identity with *R. monacensis*, the IrR/Munich strain (GenBank accession number LN794217). Among the six *Borrelia*-positive individual ticks, four were carrying *B. garinii* sequences and two *B. valaisiana* sequences, both belonging to the *B. burgdorferi* sensu lato (s.l.) complex. Regarding *B. garinii*, one sample carried sequences 100% identical to the isolate B12F6Swallowcliffe17 (acc. no. OL848409), two samples carried sequences 100% identical to the isolate B13D2Donheads15 (acc. no. OL848423), while the sequence of the fourth tick presented 99.2% identity with the isolate B13D2Donheads15. The two *B. valaisiana* sequences were identical to those of the isolate B13E4GCommon (acc. no. OL848426) [22].

Regarding the three phlebovirus-positive tick pools, all were carrying sequences belonging to the *Ixovirus* genus in the *Phenuiviridae* family, differing from the *Ixovirus norvegiae* (Norway phlebovirus 1) isolate NOR/A2/Bronnoya/2014 (acc. no. NC_055434) [23] by 1.9–2.6% and 0.7–2.1% at the nucleotide and amino acid levels, respectively (Figure 2).

Among the seven nairovirus-positive tick pools, one was clustered in the genus Orthonairovirus (*Nairoviridae* family), and six were clustered in the genus *Norwavirus* (*Nairoviridae* family). The sequence of the orthonairovirus presented 90.3% identity at nucleotide level (100% at amino acid level) to the Sulina virus (acc. no. NC_078999) detected in 2016 in *I. ricinus* ticks collected at the Danube Delta in Romania [24] (Figure 3). The six Norwavirus sequences clustered in a clade comprising the Norway nairovirus 1, Pustyn virus, and Grotenhout virus, presenting 94.74–98.03% (with a mean of 96.6%) identity with the Grotenhout virus isolate Gierle-1 detected in 2009 in *I. ricinus* ticks collected in Belgium (acc. no. NC_078265) [25] (Figure 4). At the amino acid level, the norwaviruses were 100% identical to each other and the Grotenhout virus.

## 4. Discussion

In contrast with most previous studies conducted in Greece on ticks collected from humans or livestock, *I. ricinus* was the predominant (90.45%) tick species in the present study due to the selection of a mountainous area with deciduous forest at an altitude above 600 m, representing a favorable ecosystem for this tick species, like that in temperate Europe, where this tick species is widespread. The only previous study in Greece in which *I. ricinus* predominated (44.57%), was the one conducted during specific months of the year in humid hilly areas with neighboring forests [7]. Further studies are needed in the country to validate this assertion by encompassing broader geographical regions with contrasting habitat characteristics. The fact that ticks were not found in *Pinus nigra* forested areas is in accordance with a study in Belgium, where the abundance of ticks at all stages was higher in oak stands compared with pine stands and increased with increasing shrub cover, probably related to the main hosts’ habitat preference [26]. The finding of a limited number of *I. ricinus* adult ticks in locations characterized by typical Mediterranean maquis vegetation likely resulted from animal movement between such areas and adjacent higher-altitude deciduous forests, rather than indicating true habitat suitability [5,27]. This remains to be addressed by the application of appropriate modeling approaches.

Numerous pathogens use *I. ricinus* as a vector, such as TBEV and species of *Rickettsia*, *Borrelia*, Anaplasma, Ehrlichia, and *Babesia*. Bacteria belonging to three of these genera (*Rickettsia*, *Borrelia*, and *Anaplasma*) were detected in *I. ricinus* collected in Mount Vermion, together with some recently identified phleboviruses and nairoviruses. Other pathogens were not detected, probably due to their low prevalence in the tested area. Among pathogens, *Rickettsia* spp. displayed the highest MIR and MLE values (12.81% and 18.45%, respectively). All obtained sequences were identical to the *R. monacensis* IrR/Munich isolate (acc. no. LN794217), which was identified in May 1998 in adult *I. ricinus* ticks in Munich, Germany [28]. *R. monacensis* is a member of the spotted fever group and is rarely associated with disease in humans [29,30,31]. It has a wide distribution in Europe, with a varying prevalence in *I. ricinus* ticks, from 1% in Germany to 57% in Italy (as reviewed by [32]). In a previous study in Greece, in which the ticks were collected from goats, the only *Rickettsia* species detected in *I. ricinus* ticks was *R. monacensis*. It was found in six of nine tick pools, and all sequences (the partial *Omp*A gene) were identical to that of *R. monacensis* IrR/Munich [9]. In another study on 153 ticks removed from humans during 2008–2009, when unusually increased tick aggressiveness was observed in northeastern Greece, *Rhipicephalus sanguineus* predominated (86.3%), and only two ticks were identified as *I. ricinus*. One of them was infected with *R. monacensis*, and the obtained sequences (partial *Omp*A gene) were identical to that of *R. monacensis* IrR/Munich [4]. There is not any report of human rickettsiosis associated with *R. monacenis* in Greece, where most cases of Mediterranean spotted fever are caused by *R. conorii* subspecies *conorii*, which is transmitted mainly by *R. sanguineus* [33].

*I. ricinus* also serves as a vector for various genospecies of the *B. burgdorferi* s.l. group, which are spirochete bacteria responsible for Lyme borreliosis, the most prevalent tick-borne disease in the northern hemisphere [34]. In the present study, *B. burgdorferi* s.l. was detected in almost half (45%) of the tick pools (MIR 8.79%; MLE 10.92%). However, since the ticks as a total were not tested individually, but in pools of varying sizes, it can be said that there was an identification of presence but no estimation of the exact prevalence of the pathogens. The epidemiology of Lyme disease in Greece is unknown, and although there are cases with symptoms resembling those of Lyme borrelliosis, only a few have been laboratory-confirmed (unpublished data). A probable explanation might be that few areas in Greece fulfill the criteria for a suitable habitat for *I. ricinus* with the presence of key tick hosts, while underdiagnosis cannot be excluded. In Europe, Lyme borreliosis is mainly caused by two genospecies, *B. garinii*, associated with neuroborreliosis, and *B. afzelii*, associated with erythema migrans and acrodermatitis chronic atrophicans, while *B. burgdorferi* sensu stricto, related to arthritis, is present in a few parts of Europe. Cases associated with *B. bavariensis* and *B. spielmanii* have also been reported [35]. *B. garinni* and *B. valaisiana* were detected in the sequenced specimens. These two genospecies predominated among the detected *Borrelia* spp. in a recent study in Spain [36].

*A. phagocytophilum*, an obligate intracellular bacterium causing human granulocytic anaplasmosis, was detected in two tick pools in this study (MIR 0.50%; MLE 0.51%). One was co-infection with *Borrelia* spp. Similar coinfections are often described in Europe, where both pathogens are prevalent [37]. The results of a study in Greece showed that the bacterium was detected in 4 of 45 pools (8.9%) of *I. ricinus* ticks collected from goats in northern Greece, which provided the first molecular evidence for the circulation of *A. phagocytophilum* in *I. ricinus* ticks in the country [38]. There are a few reports on human anaplasmosis in Greece, including one fatal case [39,40]. Two seroprevalence studies in the Macedonia and Crete Regions of Greece showed the presence of IgG antibodies in 7.3% and 21.4% of blood donors, respectively, suggesting that, most probably, the disease is underdiagnosed, and more awareness is needed [41,42].

The rapid development of high-throughput RNA sequencing and meta-transcriptomics has resulted in the identification of a plethora of novel microorganisms, mostly viruses, including viruses of the *Phenuiviridae* and *Nairoviridae* families. Phleboviruses are transmitted mainly by phlebotomine sandflies. However, various phleboviruses have been detected in ticks in several countries, including Greece [43,44]. For this reason, testing for phleboviruses was added to this study and resulted in three pools positive for *Ixovirus norvegiae* (Figure 2). The detected virus was initially identified in Norway using a bulk RNA-sequencing approach on six libraries consisting of 33 *I. ricinus* nymphs and adults. The virus, named Norway phlebovirus 1, was detected in all six libraries and clustered together with Blacklegged tick phlebovirus 1 [23]. Similar viruses have also been detected in Belgium, France, and Bulgaria [45,46,47].

In addition, two recently discovered nairoviruses were detected: the Sulina virus (*Orthonairovirus* genus) and the Grotenhout virus (*Norwavirus* genus) (Figure 3 and Figure 4). The Sulina virus was initially detected through next-generation sequencing in *I. ricinus* ticks collected from domestic dogs and one questing tick in the Danube Delta in Romania. Based on negative serological and cultural results, the authors suggested that the virus is associated with the virome of *I. ricinus* [24]. A similar virus was detected in France [47]. It is of interest that the closely related Yezo virus identified in 2021 in Japan is associated with febrile illness in humans [48]. The second nairovirus, the Grotenhout virus, was identified in 2009 through the next-generation sequencing of a pool of adult *I. ricinus* ticks in Belgium [25]. Similar viruses (with various names) are also present in several European countries [23,47,49]. As above, the closely related Beiji virus identified in China is associated with febrile illness in humans [50]. Unlike the typical three-segmented bunyaviral -ssRNA genome, Norwavirus genomes are bi-segmented, missing the M segment.

## 5. Conclusions

The present study showed that there is an established *I. ricinus* population in specific areas in Greece, such as Mount Vermion, which are infected with various pathogens (mainly *Rickettsia* spp.), suggesting that country-wide studies are needed, firstly, to identify the temporal and spatial distribution and abundance of *I. ricinus* ticks and secondly, to estimate the rate of ticks infected with various pathogens. Although TBEV was not detected in the present study, it should be included in the screening list of pathogens in neurological cases as there are reports of human cases in Greece. It is expected that climate change together with various environmental and anthropogenic factors will result in alterations in the distribution of vectors, including ticks, which may result in the spread of tick-borne diseases; therefore, surveillance data must be updated often to ensure the efficient design of control and prevention strategies to mitigate the risk associated with tick-borne pathogens. Regarding the detected phleboviruses and nairoviruses, further studies are needed to find out whether they are important for public and animal health or play any role in the life cycle of the ticks.

## Figures and Tables

**Figure 1 pathogens-13-00449-f001:**
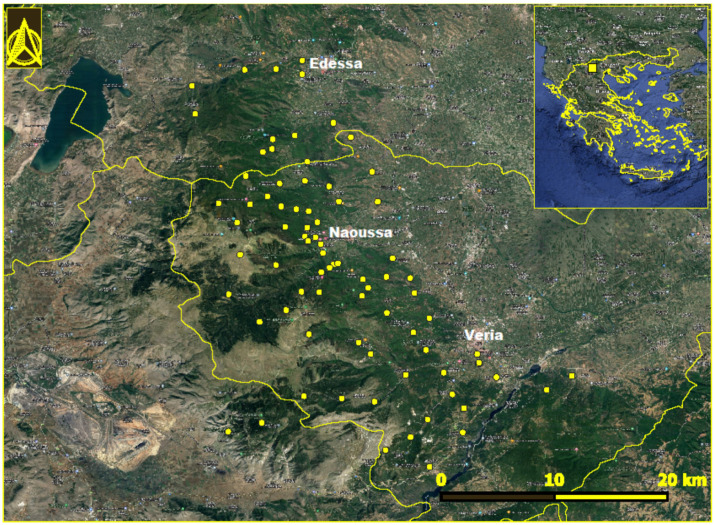
Locations in Mount Vermion, northern Greece, where *Ixodes ricinus* ticks were collected. The location of the area in Greece is seen in the inset.

**Figure 2 pathogens-13-00449-f002:**
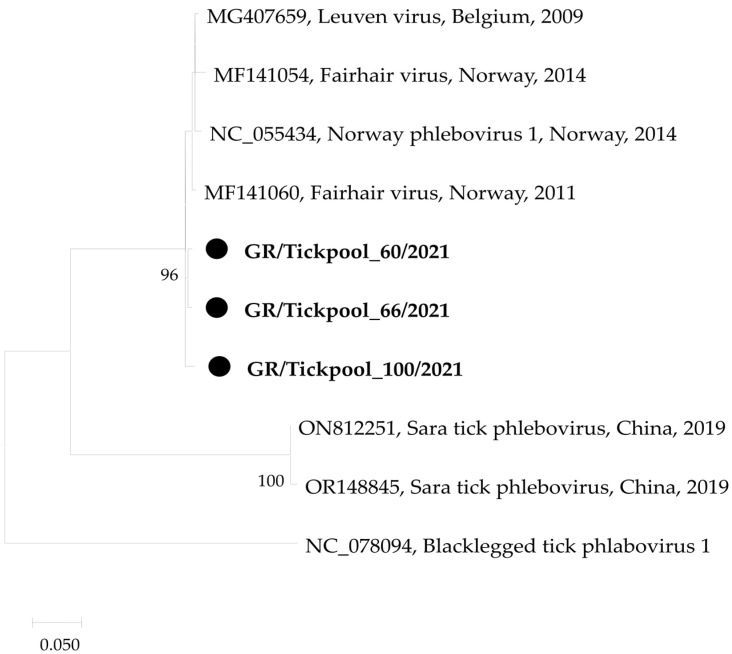
Maximum likelihood phylogenetic tree based on 479 bp fragment of polymerase gene of phleboviruses. The evolutionary distances were computed using the Kimura 2-parameter model. The percentage of replicate trees in which the associated taxa clustered together in the bootstrap test (1000 replicates) is shown next to the branches. Bootstrap values lower than 70% are not shown. The tree was drawn to scale with branch lengths measured in the number of substitutions per site. Sequences from the current study are in bold and shown with a circle.

**Figure 3 pathogens-13-00449-f003:**
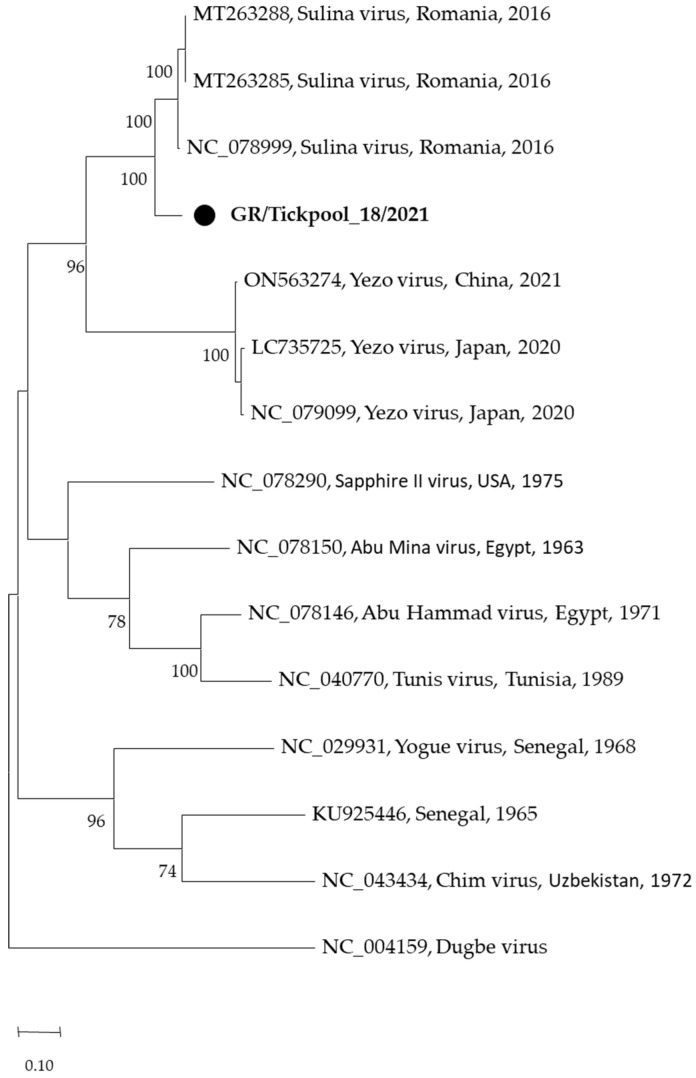
Maximum likelihood phylogenetic tree based on 495 bp fragment of polymerase gene of nairoviruses (*Orthonairovirus* genus). The evolutionary distances were computed using the Tamura 3-parameter model. The percentage of replicate trees in which the associated taxa clustered together in the bootstrap test (1000 replicates) is shown next to the branches. Bootstrap values lower than 70% are not shown. The tree was drawn to scale with branch lengths measured in the number of substitutions per site. Sequences from the current study are in bold and shown with a circle.

**Figure 4 pathogens-13-00449-f004:**
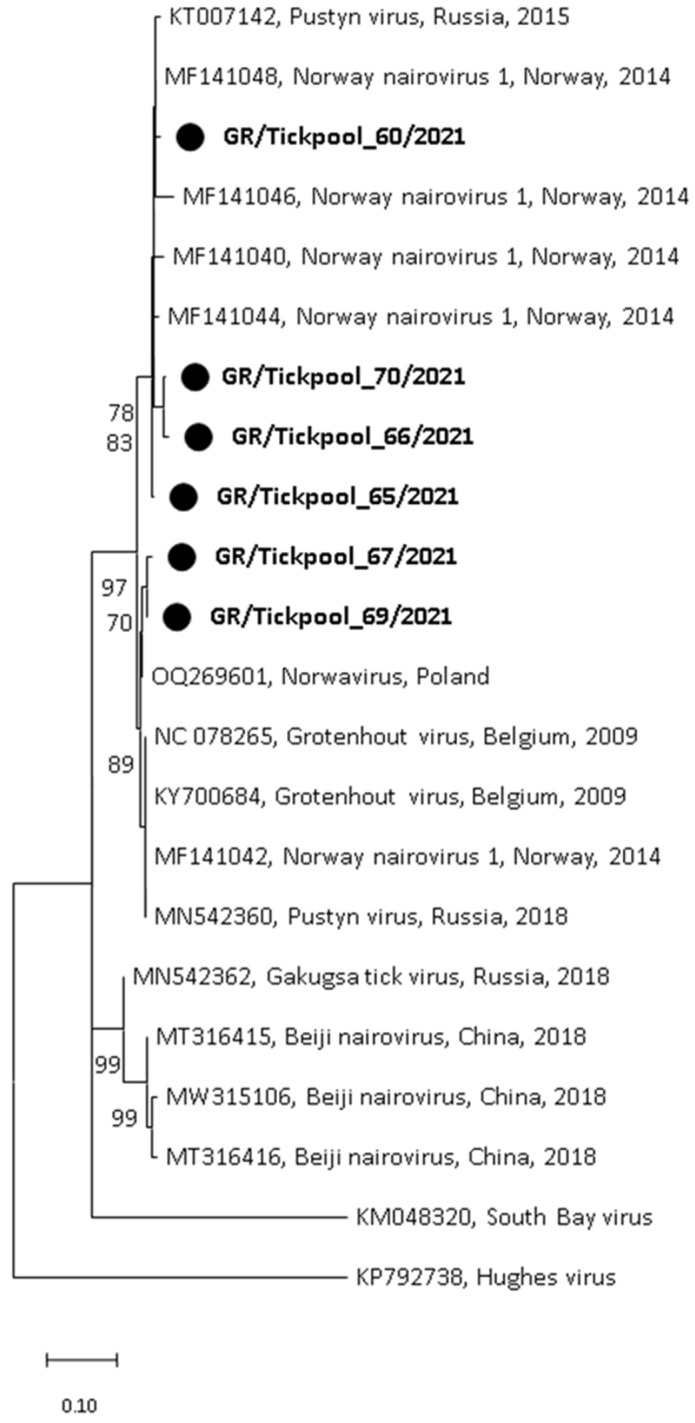
Maximum likelihood phylogenetic tree based on 154 bp fragment of polymerase gene of nairoviruses (*Norwavirus* genus). The evolutionary distances were computed using the Kimura 2-parameter model. The percentage of replicate trees in which the associated taxa clustered together in the bootstrap test (1000 replicates) is shown next to the branches. Bootstrap values lower than 70% are not shown. The tree was drawn to scale with branch lengths measured in the number of substitutions per site. Sequences from the current study are in bold and shown with a circle.

**Table 1 pathogens-13-00449-t001:** Tick species collected in Mount Vermion during April 2021–March 2022.

Tick Species	Number (%)
*Ixodes ricinus* (female)	157 (35.7)
*Ixodes ricinus* (male)	190 (43.2)
*Ixodes ricinus* (nymph)	49 (11.1)
*Ixodes ricinus* (larva)	2 (0.5)
*Dermacentor marginatus* (female)	5 (1.1)
*Dermacentor marginatus* (male)	4 (0.9)
*Haemaphysalis punctata* (female)	2 (0.5)
*Haemaphysalis punctata* (male)	1 (0.2)
*Haemaphysalis inermis* (female)	9 (2.0)
*Haemaphysalis inermis* (male)	2 (0.5)
*Haemaphysalis inermis* (nymph)	2 (0.5)
*Haemaphysalis parva* (female)	2 (0.5)
*Haemaphysalis parva* (male)	2 (0.5)
*Ixodes frontalis* (nymph)	6 (1.4)
*Ixodes gibbosus* (female)	2 (0.5)
*Rhipicephalus sanguineus* s.l. (male)	1 (0.2)
*Rhipicephalus turanicus* s.l. (female)	2 (0.5)
*Rhipicephalus turanicus* s.l. (male)	2 (0.5)
Total	440 (100)

**Table 2 pathogens-13-00449-t002:** Bacteria and viruses detected in the pools of *Ixodes ricinus* ticks collected in Mount Vermion. The number of positive pools. The MIR and MLE values with confidence intervals (CIs) are shown.

Microorganism		Positive Pools (%)	MIR% (CI)	MLE% (CI)
Bacteria	*Rickettsia* spp.	51 (63.75)	12.81 (9.53–16.10)	18.45 (14.02–23.53)
	*Borrelia* spp.	36 (45.0)	8.79 (6.01–11.58)	10.92 (7.80–14.72)
	*A. phagocytophilum*	2 (2.5)	0.50 (0.00–1.20)	0.51 (0.09–1.64)
Viruses	Phleboviruses	3 (3.75)	0.75 (0.00–3.05)	0.77 (0.20–2.05)
	Nairoviruses	7 (8.75)	1.76 (0.47–3.05)	1.82 (0.80–3.56)

**Table 3 pathogens-13-00449-t003:** Infections and co-infections of *Ixodes ricinus* ticks collected in Vermion Mountain.

Microorganism	No. of Pools (%)
*Rickettsia* spp.	22 (27.5)
*Borrelia* spp.	8 (10.0)
*Rickettsia* spp. + *Borrelia* spp.	21 (26.25)
*Rickettsia* spp. + *A. phagocytophilum*	1 (1.25)
*Borrelia* spp. + Nairovirus	1 (1.25)
*Rickettsia* spp. + *Borrelia* spp. + *A. phagocytophilum*	1 (1.25)
*Rickettsia* spp. + *Phlebovirus* + Nairovirus	1 (1.25)
*Rickettsia* spp. + *Borrelia* spp. + Phlebovirus + Nairovirus	2 (2.5)
*Rickettsia* spp. + *Borrelia* spp. + Nairovirus	3 (3.75)
Negative	20 (25.0)
Total	80 (100.0)

## Data Availability

The raw data supporting the conclusions of this article will be made available by the authors on request.

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
