# Peer review of "Pathogens Detected in Questing Ixodes ricinus Ticks in a Mountainous Area in Greece"

_pathogens, 2024, doi:10.3390/pathogens13060449_

Round 1
Reviewer 1 Report
Comments and Suggestions for Authors
The manuscript (pathogens-2989802) describes the study of Ixodes ricinus distribution in a mountainous area in Greece and their infection with a wide range of tick-borne pathogens. Previously, I. ricinus from Greece was practically not studied for the presence of tick-borne pathogens, so the purpose of this study is relevant. Unfortunately, the conducted study and submitted manuscript have many shortcomings.
1. In my opinion, the main fault of the study was that the ticks were examined not individually, but in pools. This approach can be used to detect rare pathogens (e.g., viruses), but not the common Rickettsia spp. and Borrelia spp. As a result, the prevalence of these agents in I. ricinus remained unknown. Moreover, the testing of pools of 17 ticks should not be used even for the detection of rare agents.
2. The authors constantly assessed the prevalence of agents as the percent of positive pools. This is neither correct nor informative, especially if the size of the pools varies significantly (from 1 to 17). The minimal infectious rates (MIR) should be calculated and used throughout the text.
3. Section 2.4. RNA extraction and pathogens detection. Please clarify, did you use the QIAamp cador Pathogen Mini Kit for extraction of RNA or total NA? If only RNA was extracted, how were Borrelia spp. and Rickettsia spp. amplified by in house PCR?
4. Lines 154-156. Surprisingly, only one rickettsial species, R. monacensis, was identified among 51 positive pools. Please indicate, how many rickettsial specimens have been sequenced. What was the length of sequenced fragments? To prove that your results are correct, indicate the number of nucleotide substitutions between your sequences and the other nearest rickettsial species. In Discussion section, please compare your results with known data on the prevalence of R. monacensis and other Rickettsia spp. in I. ricinus from different regions, including your own results ( 2017. doi: 10.1016/j.ttbdis.2016.09.011). Also, discuss the possibility of mixed infection of R. monacensis with other Rickettsia spp. due to analysis of pooled specimens.
5. Lines 167-163. Only 6 of 36 B. burgdorferi specimens were sequenced. These data are not representative. More B. burgdorferi samples need to be sequenced, probably using nested PCR.
6. The most interesting and novel result of the study is the finding of quite rare species of tick-transmitted viruses. This part of the text should be presented in more detail, including the description of the phylogenetic positions of identified viruses. The manuscript must also include phylogenetic trees of phleboviruses and nairoviruses. Please indicate, what was the length of the sequenced fragments?
7. Line 202. The determined MIR for B. burgdorferi is not significant (about 8%); the prevalence of B. burgdorferi cannot be determined based on obtained data.
8. Lines 207-209. In Europe, Lyme borreliosis is also caused by B. bavariensis.
Author Response
We thank the reviewers for their valuable comments and suggestions.

Reviewer 2 Report
Comments and Suggestions for Authors
Comments and Suggestions for Authors
In this study, the authors investigated the prevalence of tick-borne pathogens in Ixodes ricinus ticks collected in a mountainous area of Greece. They used a variety of molecular methods to detect a range of pathogens, including Borrelia burgdorferi sensu lato (s.l.), Rickettsia spp., Anaplasma phagocytophilum, Babesia spp. etc.
The findings of this study are important for public health, as they provide information about the prevalence of tick-borne pathogens in that area of Greece. This information can be used to develop targeted prevention and control strategies to reduce the risk of tick-borne diseases.
Overall, this is an interesting and well-conducted study that makes a valuable contribution to understanding of the epidemiology of tick-borne diseases in Northern part of Greece.
Introduction
Comment 1: The introduction would be enhanced by incorporating information about the distinction between tick collection methods, specifically the difference between collecting ticks from hosts and collecting ticks directly from vegetation. This addition would provide context for the study's methodology and highlight the potential impact of the collection method on the findings.Methods of tick collection can influence the composition and prevalence of detected pathogens. Collecting ticks from hosts, such as livestock or humans, targets ticks that have already successfully fed and are more likely to be infected. In contrast, collecting ticks directly from vegetation by dragging captures a broader range of tick life stages, including unfed nymphs/adult, which may not yet harbor pathogens.
Comment 2: line 50 ”H. punctata and H. Impeltatum” As presented, the use of abbreviations for the two hard tick genera can be confusing for readers. Therefore, I recommend using the full names of the genera without abbreviation. Using full names can help to avoid potential confusion between Hyalomma and Haemaphysalis
Materials and methods
Comment 3: Line 87-92 The provided text about the climate can be improved by citing a source for the information.
Comment 4: The manuscript would be enhanced by incorporating more detailed information about the habitats from which the ticks were collected. If available, this information could be presented in a table or another reader-friendly format. Additionally, if temperature, relative humidity, and collection area (m2) data were recorded, it would be possible to calculate tick density for each area, providing valuable insights into the spatial distribution of these organisms.
Comment 5: line 105 Pinus nigra à Pinus nigra (Italic)
Comment 6: Recommendation for clarifying the origin of Figure 1
The article would benefit from a clarification regarding the origin of Figure 1, the map depicting the study area. Currently, there is no indication of whether the map was created using a specific software or if it is based on an existing source. Providing this information would enhance the transparency and credibility of the study's presentation.
Comment 7: Regarding the Materials and Methods section, it would be beneficial to include information on how exactly the ticks from which nucleic acids (DNA/RNA) were isolated were grouped in pools. As presented, the criteria are not clear. Additionally, it is confusing how the ticks were grouped and tested in a pool, but then subsequently examined individually (line 127-132).
Comment 8: line 114: -80oC correct to -80oC
Comment 9: line 115 and 118. Please clarify whether RNA or DNA was isolated from the ticks. It is likely a technical error. So, please review and correct the text if necessary.
Comment 10: I believe a citation is needed regarding point 2.5 for the Sanger sequencing and analysis.
Results
Comment 11: The tick pathogen detection kit you are using offers the ability to detect a wide range of pathogens. Please also include negative results. From an epidemiological point of view, it is also important whether you have reported the presence of Relapsing fever Borrelia group, Coxiella burnetii, and Babesia which are not mentioned in the results and/or discussion section.
Comment 12: line 147 correct 295m to 295 m (space) and 600m to 600 m (space). The same for line 185 (600 m)
Discussion
Comment 13: On line 195, the sentence starts with an abbreviation. Since it is the beginning of a sentence, it is not appropriate to start with an abbreviation. Please change the sentence to an appropriate form.
Comment 14: I had the pleasure of reviewing the submitted article. I believe it presents valuable research that contributes to our understanding of the role of Ixodes ricinus as a vector of various pathogens. However, I would recommend expanding the discussion section. The study findings provide a rich foundation for a more in-depth discussion that could significantly enrich the article.
With regards!
Author Response
We thank the reviewers for their valuable comments and suggestions!

Round 2
Reviewer 1 Report
Comments and Suggestions for Authors
The authors of the manuscript pathogens-2989802 significantly improved their manuscript during revision. All necessary details were added and the limitations of the study were stated.
I only have two minor comments:
1. In ”Abstract”, the number of examined ticks should be specified.
2. Line 263. “The bacteria belonging to three of these genera” should be instead of “Three of these bacteria”
Despite this, I am not satisfied with the quality of the study described in this manuscript. Indeed, the authors first examined questing I. ricinus from Greece for tick-borne pathogens. Rickettsia spp. and Borrelia spp. were found in 51 and 35 tick pools, respectively. However, only 6 rickettsial and 6 borrelial specimens have been genetically characterized to the species level. Thus, the species and genetic diversity of rickettsial and borrelial agents in I. ricinus in Greece is not established. Notably, all viral agents were successfully sequenced.
I strongly encourage authors to use nested PCR-based sequencing in their future studies for genetic characterization of bacterial agents.
Author Response
We thank once again Reviewer 1 for all comments!
- The number of ticks is included in the abstract.
- Line 263. We replace the words with the suggested and we named them "Bacteria belonging to three of these genera (Rickettsia, Borrelia, Anaplasma)".
We promise to use nested PCRs for typing Rickettsia and Borrelia in our future studies!